# GENERATIVE AI IN HEALTHCARE: A TRUSTWORTHY APPROACH

## ABSTRACT

The recent advancements in self-supervised algorithms like Transformer Architecture and Diffusion models have expanded the means of applying AI in healthcare and life sciences. To achieve real-world adoption, it is important to measure and audit the trustworthiness of the AI system as per the legal and compliance requirements for privacy, security, fairness, and safety. In this paper, we focus on the method to achieve trustworthiness in an LLM (Large Language Model) based decision support system for physicians. The stakeholders for this decision support system are patients, physicians, regulators, and external auditors. We focus on the limitations of large or foundation models and the method to overcome these limitations, with the aim of accelerating the adoption of this far-reaching technology in the healthcare sector. It also explores possible guardrails for safety and the methods for aligning AI systems to guardrails.

## 1 OUR SOLUTION APPROACH

We explore an approach to an AI system that can enhance decision capabilities by using the data and EHRs (Electronic Health Records) collected over many years for a vast volume of patients. The longitudinal data consists of clinical details, biomarkers, disease progression indicators, treatment administered, and patient outcomes. The goal of the system is to assist physicians in identifying the best treatment option for a given patient context. The LLM-based system will be able to predict optimal options based on hundreds of similar cases on which it was trained. The paper addresses the transparency, data integrity, model development, and performance validation of the system. In the sections below, we explore the various stages of development and deployment of such a system, the challenges, and the methods to overcome the challenges.

### 1.1 AUGMENTING HUMANS WITH ACCOUNTABILITY AND TRANSPARENCY

The AI systems, in the near future, will augment human capability rather than replace humans. Conformity and ongoing compliance with legal codes and ethical values can be achieved by a human-led governance mechanism. The usage of AI needs to be based on applicable regulatory guidelines for critical, serious, and non-serious use cases. As the technology evolves, the guidelines issued by the regulatory agencies are also evolving. There is no single guideline or protocol that can be considered comprehensive and complete. Regulatory frameworks like FDA's SaMD are valuable but additional guardrails specific to large and multi-models will be necessary as large model-specific challenges like hallucinations are not addressed.

The AI-based physician decision support solution provides a mechanism to query for diagnosis and effective treatment as per a patient's context and delivers insights that are inferred based on training done on high-volume historical data. The method that we are suggesting is human-led accountability, where the options are generated via AI but the final decision is taken by a qualified physician. This solution brings the knowledge within the data generated by the hospital over many years to each physician's fingertips and enables them to make the best possible decision.

Large language models cannot determine treatment decisions directly. They can present unbiased information about knowledge gained from similar cases to clinicians and augment clinicians' decision-making capacity. This feature can especially be important in special case scenarios where guidelines alone cannot direct the decision. For instance, in a co-morbid ascitic patient with nephrotic syn-

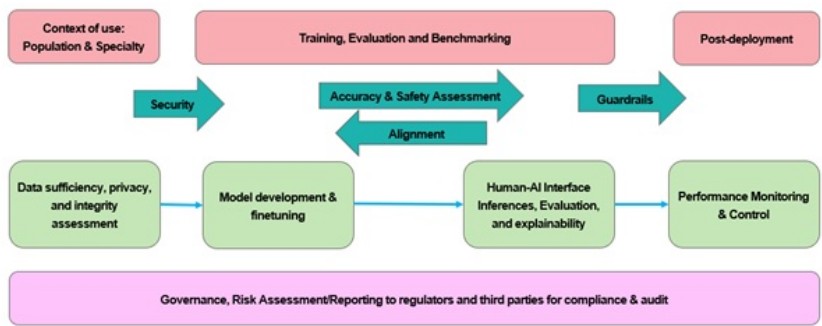

Figure 1: AI-based physician decision support solution. Lifecycle workflow for data evaluation, model evaluation, inference validation, and monitoring of LLM-based AI model for healthcare applications

drome and liver disease, how much fluid should be removed by an ascitic tap? Clear guidelines in such cases are unavailable and the decision mainly lies with the clinician. However, having detailed information about similar cases on hand will be a helpful guide to the clinician's decision-making process.

## 2 DATA SUFFICIENCY AND INTEGRITY ASSESSMENT

Completeness, representativeness (fitness for purpose), and accuracy of the data need to be validated and the validation process should answer the following questions.

Has the defined context of use, such as specific populations, age groups, sex, and comorbidities, been recorded? Is the data accurate? Has the process followed in collecting and consolidation been designed as per regulatory guidelines? Does the data represent the context of use? Have the longitudinal aspects of data, throughout the treatment trajectory and for different contexts, been captured?

Privacy and security-related questions: Are there protocols and audits in place for the protection and privacy of data as per legal requirements? For example, is it compliant with HIPPA (Health Insurance Portability and Accountability Act)?

Data used from the EHR system must be verified by a clinician before being fed into the system. Adequate representation to ensure the similarity of the training dataset to the general population where the system is deployed should be ascertained and verified. Including outpatient data can prevent berksonian bias.

Data for each case scenario should be captured for all possible contexts. And when providing suggestions contexts of use should be established, and information pertaining to the context of use should be provided. For instance, a patient with diabetes, hypertension, and coronary heart disease is being evaluated for CABG. The patient can be evaluated from the context of cardiovascular as well as endocrine pathology.

## 3 MODEL DEVELOPMENT AND FINETUNING

Foundation models are multi-purpose with the same model being used for a range of functions. A model trained on EHR and disease progression can be used for predicting the best treatment option as well as for recording the statistics about the patients who followed different treatment regimes. The same model can be used to classify the disease stage. A foundation model trained with multiple modalities of data with cross-attention is very suitable for applications in health and life sciences.

The challenges of foundation models and our approach to address them are explained below.

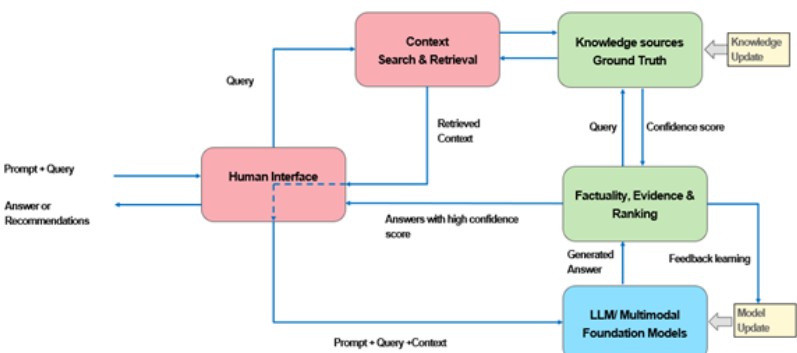

Figure 2: Retrieval-augmented generation & retrieval-augmented confidence score for factuality. Overall architecture diagram showing components: User query used to retrieve the context, enriches the Context Window with referable information, method for estimating the factuality of generated information, evidence presentation, ranking of the answers, and the mechanism for feedback collection for continuous monitoring and model improvement.

## 3.1 HALLUCINATION AND ALIGNMENT

Hallucination (factually incorrect/imaginary answers) is an expected property in large models as the model's outcome is more about "the answer/prediction looks like…" rather than "the answer is…". The model output is not based on the retrieval of information from a reliable source. It is based on the best contextual access of represented knowledge by vector/tensor similarity. So, the LLM/Foundation model occasionally outputs factually incorrect answers.

In healthcare application use cases like the recommendation of a specific treatment for a patient, getting a dependably accurate answer is critical. Even though there are multiple mechanisms for alignment, none of the methods are foolproof for application in healthcare use cases. In certain applications like drug discovery (e.g., designing a protein that doesn't exist), hallucinations turn out to be useful, as the answers need to be a novel discovery that is different from the training data.

We ensure factuality/correctness with the following methods:

## 3.2 REASONING

Our method proposes a task-level (steps of a larger process) supervision approach instead of end-to-end outcome supervision for improving the reasoning and overall accuracy. The task-based approach is done step-by-step. This method rewards the steps (task supervision) instead of the overall outcome (outcome supervision).

Achieving the overall goal by integrating multiple small tasks is a better approach, as tasks/skills can be reusable for multiple larger processes. By introducing a chain of thoughts (CoT) approach, it is possible to make the methodology similar to the "System 2" concept. Such an approach helps in defining and optimizing the objective functions of the algorithms.

As an additional benefit, reasoning helps in achieving explainability. Each step in the sequence and the overall outcome contribute to the explanation which is an essential criterion for any healthcare AI application. EFor example, if a diagnosis of pneumonia is made by the AI solution, an explanation in the form of EHR inputs and the CT scan findings should be available.

Introducing a chain of thought approach will improve the explainability of the algorithm as the overall result will be determined by various modalities fed into the system. For instance, in a case of suspected lung cancer: smoking history, number of pack-years, family history, chest X-ray, CT scan, biomarkers, etc., all provide essential information. A stepwise approach focusing on each modality will help the LLM develop better explainability for its decision.

### 3.3 FINETUNING

Finetuning helps large models to specialize in a specific domain like medicine/life sciences. Introducing domain-specific data, domain-specific tokenization, supervised finetuning, and human-assisted reward modeling helps the LLM to learn and represent domain-specific knowledge. Better representation of domain-specific knowledge results in better "understanding" and, consequently, improved accuracy. Optimized mechanisms like PEFT (Parameter Efficient Finetuning) can be used for cost-efficient finetuning.

## 4 INFERENCES AND EVALUATION

### 4.1 PROMPTING

Prompting is a method to align LLM/foundation models to the context of use. Prompting is a method to increase the probability of correctness by providing a sufficient number of in-context examples, which align with the expected outcome for a given interaction. Model parameters are not changed during prompting/prompt engineering. Prompting isn't a measurable optimization and involving human domain experts will maximize the alignment with end-user preferences and accuracy. An interface for human experts to experiment and hone the prompt-engineering techniques will help in achieving the best outcomes. Along with prompting, other methods like "guidance" can be used to format the results, as per the requirements of the healthcare workflow.

### 4.2 TOOLS FUNCTION INTEGRATION

LLMs and foundation models are good at language-understanding and determining the context, but they are not good at elaborate reasoning skills like science experiments, math, and planning tasks. This limitation can be addressed by integrating APIs/functions for specialized operations (calculator)and factuality verification (medical SOP).

### 4.3 PROPOSED GUARDRAILS

This checkpoint step consists of alignment with the ground truth and minimization of uncertainty. One of the methods for alignment is crosschecking the predictions with ground truth case studies. The hallucinated/imagined predictions are filtered out, aligning the final set of outcomes to the ground truth. The recommendations are also checked for alignment with the standard operating procedure (SOP) and against the overall statistics of patient outcomes in different types of treatment. The factuality check is achieved by augmenting the "memory only" context of the foundation model with retrieval capabilities for factual correctness. Retrieval-augmented LLMs are much more reliable for factuality as now there is a mechanism/orchestration for sampling the right answers from a set of possible answers.

### 4.4 EXTRAPOLATION AND CONFIDENCE LEVEL

It is important for the system to provide reasoning about the prediction done by the algorithm. Deriving a confidence level for the prediction is challenging in large models as the probability of getting the right answer decreases with the increase in the length of the predicted tokens. LLMs are known to make wrong predictions with high confidence levels. So, the probabilistic approach for deriving the confidence score is not useful. Instead in our proposed solution, we suggest optimization of inference time objectives as per the workflow/process requirement. Accordingly, the system informs the user when the context is new, or there is no existing case study in the training data, and refrains from giving any recommendations/answers.

One additional guardrail for increasing the confidence level would be to make the system cite the top five studies it considered when analyzing the data and providing recommendations. This would enable the clinician to independently analyze the sources in case of any discrepancies. An example of an emerging disease with no prior context would be COVID-19, especially in the initial stages. Such instances are ripe for misinformation to spread, so the system should have specific guardrails in place for public health hazards which can be prone to misinformation.

### 4.5 REINFORCEMENT LEARNING WITH HUMAN-MACHINE INTERACTIONS

Post-deployment feedback helps in improving reliability and accuracy. This can be achieved by designing an effective interface for human-machine interaction where human-provided feedback is collected together with data on system performance and accuracy. A policy optimization with the right reward will further improve the system.

To derive the best possible feedback, we propose an interface where physicians can do a relative ranking of the three top answers rather than an absolute ranking. This will help in generalizing the score assigned to an answer by different individual experts. The data generated will be used for Reward Modeling (RM) or Reinforcement Learning from Human Feedback (RLHF).

## 5 MONITORING AND CONTROL

Explainability, Performance validation and monitoring:

Explainability: As the mathematical interpretability of the LLMs is still a challenge, it is necessary to have concept-level explainability. Explanation needs to be done at the process and task level. Process-level explainability is achieved by Chain of Thoughts (CoT). The task/skill level explainability can be achieved by Testing with Concept Activation Vectors (CAV), which verifies the prediction based on conceptual matching which is much beyond mere pattern matching.

Safeguarding against data drift and data leaks during test/evaluation time and post-deployment is very important. Data leak occurs when the test samples are very similar to training samples.

Retrieval-augmented LLMs may also have a mechanism for constantly updating the ground truth according to the latest available literature since medicine is a constantly evolving field. A workflow similar to the medical encyclopedia UpToDate could be employed, with relevant field clinicians being charged with updating the knowledge base on a timely basis. These instructions apply to everyone, regardless of the formatter being used.

## 6 CONCLUSION

Generative AI has great potential if used carefully, with necessary data governance and factuality validation of the generated output for the context of use. A completely deterministic system is not possible with transformer-based architecture, but the probabilistic solution is powerful in providing dependable information to physicians/clinicians. A contextual and conversational approach further enhances the utility of the solution. Privacy, fairness, and safety-related guardrails are important aspects of the application of generative AI in healthcare. These aspects can be achieved in the solution by applying the proposed approach in the paper. We propose a close collaboration between AI experts and healthcare domain experts for an optimal outcome. Further progress in understanding the LLM/Foundation model will help in improving the factuality of generative models.

## 7 CITATIONS, REFERENCES

Hu et al. (2021) Panigrahi et al. (2023) Shevlane et al. (2023) Zhou et al. (2022) Lanham et al. (2023) Kahneman (2011)

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
