# OpenReview forum: "Generative AI in healthcare: A trustworthy approach"
_ICLR.cc/2024/Conference — ICLR 2024 Conference Withdrawn Submission_

### Official Review · Reviewer_8VP9 · 2023-10-24

**Soundness:** 1 poor
**Presentation:** 1 poor
**Contribution:** 1 poor
**Rating:** 1
**Confidence:** 5

**Summary:**

This paper explores the potential of LLM in healthcare AI systems and emphasizes the importance of trustworthiness and safety in such scenarios. It discusses the limitations of large or foundation models in healthcare and highlights the need for aligning AI systems with guardrails for safety in healthcare. Overall, the paper proposes a general discussion on the role of generative AI and LLM in healthcare by providing qualitative insights on how these models can support physicians in identifying the best treatment option for a given patient context.

**Strengths:**

The paper addresses an important and timely topic - the use of generative AI and LLM in healthcare - and highlights the need for trustworthiness and safety in such systems.

**Weaknesses:**

The paper does not provide any methodological advance, empirical evidence or results. The article summarises qualitatively the challenges and impact of LLM for predicting the best treatment regimes in the healthcare context. Moreover, the paper does not discuss the potential ethical implications of using AI in healthcare, such as bias, privacy, and informed consent issues.
In the end, the paper does not provide any scientific contribution. It does not address the potential challenges or limitations of implementing generative AI and LLM frameworks in real-world healthcare settings.

**Questions:**

The paper has a shallow scientific quality and lacks scientific contribution.
At this stage, No response to my question can change my opinion.

---

### Official Review · Reviewer_mU33 · 2023-10-29

**Soundness:** 1 poor
**Presentation:** 1 poor
**Contribution:** 1 poor
**Rating:** 1
**Confidence:** 3

**Summary:**

The goal of this paper is to introduce methods for achieving trustworthiness in LLM-based decision support systems in healthcare. The authors lay out comprehensive topics that need to be addressed for the safe deployment of decision support systems.

**Strengths:**

- The motivation for the paper is convincing, highlighting the necessary topics to be discussed to ensure trustworthiness in LLM-based decision support systems.
- The authors provide a series of relevant questions and concerns that should be addressed for each topic.

**Weaknesses:**

- The paper seems incomplete. Some paragraphs lack in detail and the citations are not properly done.
- The paper contains numerous incorrect statements. For example, in the abstract, it mentions "self-supervised algorithms like Transformer Architecture and Diffusion models." This is a mix-up of concepts. Self-supervised learning is a type of learning objective, whereas Transformer and Diffusion models are specific types of model architectures. The parameters of these architectures are optimized based on whichever learning objective is chosen.
- The manuscript would be significantly improved if the authors correct the false claims and provided a more detailed explanation for each section.

**Questions:**

The paper raises many questions due to its lack of detail and insufficient citations. I suggest that the authors supplement the details of the paper before initiating a fruitful discussion through the OpenReview platform.

---

### Official Review · Reviewer_tPic · 2023-10-31

**Soundness:** 1 poor
**Presentation:** 2 fair
**Contribution:** 1 poor
**Rating:** 1
**Confidence:** 5

**Summary:**

The authors reported on an approach for using generative AI in healthcare. They described several steps that can be used to reliably use GenAI in the context of an AI assistant for healthcare.

**Strengths:**

The main strength of the paper lies in the context of addressing a timely topic with the looming usage of GenAI for health.

**Weaknesses:**

The main draw back of the paper is that scientific contribution in the paper is lacking. In its current form the paper is a read out of how one "could" use GenAI in health. However, there are no testable scientific hypothesis that can be reviewed and commented upon. The framework is neither justified by empirical or theoretical evidence. Even the desiderata reads like an opinion rather than one derived through expert studies or critique of past work. Overall in its current state the paper is not suitable for publication.

**Questions:**

None. see weakness above

---

### Meta-Review · Area_Chair_wrRC · 2023-12-02

**Metareview:**

I concur with the reviewers' comments. I believe their insights will help the authors to improve their paper and identify a suitable publication venue for their work.

**Justification For Why Not Higher Score:**

NA

**Justification For Why Not Lower Score:**

NA

---

### Decision · Program_Chairs · 2024-01-16

Reject